# Deep Learning for Automated Measurement of Patellofemoral Anatomic Landmarks

**DOI:** 10.3390/bioengineering10070815

**Published:** 2023-07-08

**Authors:** Zelong Liu, Alexander Zhou, Valentin Fauveau, Justine Lee, Philip Marcadis, Zahi A. Fayad, Jimmy J. Chan, James Gladstone, Xueyan Mei, Mingqian Huang

**Affiliations:** 1BioMedical Engineering and Imaging Institute, Icahn School of Medicine at Mount Sinai, New York, NY 10029, USA; 2Department of Diagnostic, Molecular, and Interventional Radiology, Icahn School of Medicine at Mount Sinai, New York, NY 10029, USA; 3Department of Orthopedics and Orthopedic Surgery, Icahn School of Medicine at Mount Sinai, New York, NY 10029, USA

**Keywords:** deep learning, body weights and measures, patellofemoral joint, arthroplasty, knee replacement, CT

## Abstract

Background: Patellofemoral anatomy has not been well characterized. Applying deep learning to automatically measure knee anatomy can provide a better understanding of anatomy, which can be a key factor in improving outcomes. Methods: 483 total patients with knee CT imaging (April 2017–May 2022) from 6 centers were selected from a cohort scheduled for knee arthroplasty and a cohort with healthy knee anatomy. A total of 7 patellofemoral landmarks were annotated on 14,652 images and approved by a senior musculoskeletal radiologist. A two-stage deep learning model was trained to predict landmark coordinates using a modified ResNet50 architecture initialized with self-supervised learning pretrained weights on RadImageNet. Landmark predictions were evaluated with mean absolute error, and derived patellofemoral measurements were analyzed with Bland–Altman plots. Statistical significance of measurements was assessed by paired *t*-tests. Results: Mean absolute error between predicted and ground truth landmark coordinates was 0.20/0.26 cm in the healthy/arthroplasty cohort. Four knee parameters were calculated, including transepicondylar axis length, transepicondylar-posterior femur axis angle, trochlear medial asymmetry, and sulcus angle. There were no statistically significant parameter differences (*p* > 0.05) between predicted and ground truth measurements in both cohorts, except for the healthy cohort sulcus angle. Conclusion: Our model accurately identifies key trochlear landmarks with ~0.20–0.26 cm accuracy and produces human-comparable measurements on both healthy and pathological knees. This work represents the first deep learning regression model for automated patellofemoral annotation trained on both physiologic and pathologic CT imaging at this scale. This novel model can enhance our ability to analyze the anatomy of the patellofemoral compartment at scale.

## 1. Introduction

Patellofemoral joint stability is a complex problem with various bony and soft tissue contributors. Substantial research in the last decade has focused on elucidating patellofemoral anatomic parameters. One area of focus is the characterization of the shape, position, and orientation of the trochlear groove, which serves as an important anatomical feature and key determinant of patellar positioning during knee range of motion [1,2,3,4,5]. Related work demonstrated differences in trochlear groove anatomy by gender [6,7] and ethnicity [7]. Other work focused on characterizing parameters of dysplastic patellofemoral anatomy compared to normal anatomy [8] and investigating the ability of knee replacement implants to replicate physiologic patellofemoral anatomy both in vivo and ex vivo [9,10,11,12,13,14]. Improving our understanding of physiological patellofemoral anatomy in populations could improve patient outcomes by reducing postoperative complications in total knee arthroplasty (TKA) and other knee arthroplasty (KA) procedures. However, studies are often limited by the availability of annotated cross-sectional imaging data or smaller sample sizes.

There is a clear clinical interest in further understanding the anatomy of the patellofemoral compartment. This often relies on the identification of seven key anatomical landmarks within the patellofemoral compartment that enable the measurement of various parameters that quantify morphological characteristics [15,16] (Figure 1). Developing a tool that enables the automated annotation of these landmarks would greatly enable the measurement of patellofemoral parameters at scale in a precise, reproducible, and time-sensitive fashion free of reader inter/intra variability.

Over the last decade, deep learning, a subset of artificial intelligence, has demonstrated remarkable utility in detecting landmarks and objects in images, such as the detection of facial landmarks for facial recognition [17,18]. These techniques can also be applied to medical imaging, particularly for patellofemoral characterization. Several studies have investigated the potential of deep learning for automated measurement of various knee parameters. For instance, E et al. developed a deep learning algorithm for automatically detecting patellofemoral joint landmarks using axial radiography in the Laurin view [19], while another study by Ye et al. demonstrated the accuracy and utility of automated patellar height measurement using deep learning [20]. Adjacent work by Simon et al. and Liu et al. developed deep learning models for automated long-leg radiograph measurements [21,22], while Kim et al. developed a deep learning model to identify loosening TKA implants from plain radiographs [23].

The success of these studies suggests that deep learning can be practically applied to study patellofemoral and trochlear groove geometry. While neural networks typically require large imaging datasets to achieve good performance, recent work by Mei et al. in developing RadImageNet, a large-scale dataset consisting of 1.35 million multimodal radiologic images, demonstrated superior performance on smaller datasets in medical imaging applications using transfer learning on RadImageNet pretrained weights [24]. Furthermore, a simple framework for contrastive learning (SimCLR) can learn image representations from self-supervised data, thereby boosting deep learning performance using pretrained weights [25]. These techniques may further enhance the accuracy and utility of deep learning for patellofemoral measurement.

We hypothesized that a deep learning model with self-supervised pretrained weights could be trained to automatically detect patellofemoral anatomic landmarks. This would demonstrate the feasibility of utilizing deep learning for measurement of key patellofemoral parameters.

## 2. Materials and Methods

### 2.1. Data Collection

This is an IRB-approved retrospective study conducted at the Mount Sinai Health System. Computed tomography (CT) images of the knee from 483 patients were acquired between April 2017 through May 2022 and used in this study. Patients from 6 sites throughout the health system were scanned with axial CT in the supine position with the knee extended. Studies were acquired with thickness ranging from 0.625–5.0 mm on CT systems from 4 manufacturers (GE, Siemens, TOSHIBA, and Philips) at 120 kVp.

Two cohorts were selected in order to improve model generalizability and performance on both physiological and pathological patellofemoral anatomy (Figure 2). In the first cohort, 447 patients with knee imaging acquired between 10 April 2017 and 7 May 2021 were evaluated. Exclusion criteria included prior instrumentation, hardware, trauma, extensive degenerative changes, or poor radiologic/reconstruction quality, leaving 206 asymptomatic patients for analysis (the “healthy cohort”). In the second cohort, 723 patients scheduled for Mako TKA procedures due to osteoarthritis (OA)-associated degeneration with preoperative CT scans acquired between 6 February 2018 and 24 May 2022 were evaluated (the “KA cohort”), and 446 patients were randomly excluded to balance cohort training sizes with the healthy cohort. In both cohorts, some studies included bilateral knee imaging and were split into 2 separate image sets. General patient cohort characteristics are summarized in Table 1.

The total 483-patient cohort was divided into the training set (329 patients: 364 knees, 9884 images), validation set (59 patients: 64 knees, 1896 images), and test set (95 patients: 105 knees, 2872 images). Patients from both the healthy and KA cohorts were randomly distributed between the training, validation, and test sets such that the overall ratio of healthy to KA patients was maintained in each set.

### 2.2. Image Collection and Annotation

The first step of image collection was the gross selection and review of the CT slice range containing trochlear groove geometry in each patient study by a PGY-4 radiological resident. Trainees (with experience ranging from first-year medical school through PGY-4 radiological residents) then annotated the 7 patellofemoral anatomic landmarks (Figure 1) on every CT image containing the trochlear groove using the Mount Sinai BioMedical Engineering and Imaging Institute’s Discovery Viewer, a PHI-protective web-based annotation tool. The senior musculoskeletal radiologist reviewed all annotations and corrected any as needed for final approval. CT images were excluded if there was no well-defined trochlear groove. Studies with bilateral knee imaging were split into 2 separate annotated images of each knee. Ultimately, 3162 CT images from the healthy cohort and 11,490 CT images from the KA cohort were annotated and subsequently preprocessed.

### 2.3. Data Preprocessing for Training

All CT images were preprocessed with a window level of 350 and window width of 2000 and then pixel normalized with a range of 0 to 255. The bony region of each CT slice was identified using a Gaussian filter with a pixel threshold of 105 and then cropped with a bounding box and reshaped into a 160 × 160 pixel image. Coordinates of each annotated landmark were computationally maintained in their relative locations during this process. These preprocessed 160 × 160 annotated images were then used for the training and development of the “aligner” deep learning model, which roughly predicts the 7 landmarks on an entire CT image.

The “patch” model further refines the predictions made by the aligner model. For this model, each 160 × 160 image was further cropped into 7 36 × 36 patch images centered around each landmark annotation with a random horizontal/vertical shift up to 10 pixels. The patch images and their relative annotation coordinates were then used as the inputs and ground truth for the training and development of 7 individual patch models for each patellofemoral landmark.

### 2.4. Deep Learning Model

In this study, a 2-stage deep learning regression model, consisting of an “aligner” model and a “patch” model, was developed to predict the coordinates of the 7 patellofemoral landmarks (Figure 3). The aligner model attempts to predict all 7 landmark locations simultaneously using a preprocessed 160 × 160 image. The aligner model yields initial predictions. Then, 7 individual patch models for each particular landmark were developed to further refine the corresponding predictions made by the aligner model. Each patch model predicts the coordinates of its specific patellofemoral landmark within a 36 × 36 patch centered around the initial coordinate predicted by the aligner model.

A modified ResNet50 architecture was used as the backbone for both the aligner and patch models, with an additional supervision mechanism connecting the last layer of each residual block to the final flattened layer. This supervision mechanism preserves the image features extracted at each residual block. The aligner model was trained on the set of 160 × 160 preprocessed full images with the relative coordinates of all 7 landmark annotations, while each individual patch model was trained on the set of 36 × 36 preprocessed patch images with the relative coordinates of the respective landmark. All models were trained using a mean squared error loss function and an Adam optimizer and initialized with self-supervised learning (SSL) pretrained weights. The model was fine-tuned with learning rates of 0.001, 0.0005, and 0.0001, and the learning rate had an exponential learning rate schedule of 0.95. All implementation and training was carried out in PyTorch (release 1.11.0 with CUDA 11.3) in Python (release 3.8.5).

### 2.5. Self-Supervised Learning Pretrained Weights

The SSL pretrained weights were obtained by training a SimCLR framework [25] with a ResNet50 backbone on the training dataset of preprocessed full-knee images. The ResNet50 architecture was initialized with RadImageNet pretrained weights [24], and the SimCLR model was trained by generating augmented image pairs and minimizing the contrastive loss. The resulting pretrained weights encapsulated valuable feature representations and served as the basis for downstream training of the aligner and patch models.

### 2.6. Statistics

Mean absolute error was calculated to evaluate the difference between the predicted coordinates and the ground truth coordinates. A paired *t*-test was used to compare patellofemoral measurements between the prediction and ground truth, while an independent *t*-test was used to compare differences in patellofemoral measurements between the healthy and KA cohorts. A *p*-value less than 0.05 was considered statistically significant. Analyses were performed using the statsmodels package (release 0.13.2) in Python (release 3.8.5).

## 3. Results

### 3.1. R1: Landmark Detection by Deep Learning Models

The spatial accuracy of the predicted landmarks was critical for the evaluation of model performance, as all patellofemoral parameters are calculated based on the predicted coordinates. The performance of the patch-based model was evaluated by comparing the coordinates of the predicted patellofemoral landmarks with the ground truth annotation coordinates. The mean absolute error (i.e., spatial distance) between the prediction and the ground truth landmarks was 0.20 cm in the healthy cohort and 0.26 cm in the KA cohort at the original 512 × 512 image resolution. This indicated highly accurate automated landmark detection comparable to human readers. Examples of predictions made by the patch-based model are presented in Figure 4.

In our two-stage model, we obtain landmark predictions from the aligner model and the patch models. In Table 2, we compare the performance of the aligner model and patch models by evaluating the average Euclidean distance between the predicted and ground truth landmark coordinates on the test dataset. When using 0.40 cm as the distance threshold, the patch models had more precise predictions, with a smaller average Euclidean distance, and higher accuracy on both healthy and KA cohorts compared to the aligner model.

### 3.2. R2: Measurements of Patellofemoral Parameters

Based on the landmarks predicted by the patch models, the following patellofemoral parameters were measured: transepicondylar axis (TEA) length, TEA-posterior femur axis (PFA) angle, trochlear medial asymmetry ratio (the ratio of M–L/G–L), and sulcus angle (Figure 1). A summary of all predicted patellofemoral measurements is shown in Figure 5 and Table 3.

Transepicondylar length was measured as the distance between the most medial point and the most lateral point of the predicted landmarks on all slices of each study. The slice containing the longest transepicondylar length was then used to define the TEA. In the healthy cohort, the mean TEA length predicted by the deep learning model was 8.16 cm, while the ground truth mean TEA length was 8.17 cm. Similarly, in the KA cohort, the mean predicted TEA length was 8.06 cm, while the ground truth mean TEA length was 8.07 cm. There was no statistically significant difference between the ground truth and deep learning model predictions of TEA length in both the healthy and KA cohorts (*p* = 0.55 and 0.30, respectively). Additionally, there was no statistically significant difference in the ground truth TEA lengths between the healthy and KA cohorts (*p* = 0.41).

After the epicondyle landmarks and TEA were identified, the PFA was defined by connecting the most posterior medial and lateral landmarks of the distal femur. Then, the angle between the TEA and PFA was measured. In the healthy cohort, the mean predicted TEA-PFA angle was 5.87°, while the mean ground truth TEA-PFA angle was 5.96°. Similarly, in the KA cohort, the mean predicted TEA-PFA angle was 6.36°, while the mean ground truth TEA-PFA angle was 5.44°. There was no statistically significant difference between the ground truth and deep learning model predictions of TEA-PFA angle in both the healthy and KA cohorts (*p* = 0.17 and 0.76, respectively). Additionally, there was no statistically significant difference in the ground truth TEA-PFA angles between the healthy and KA cohorts (*p* = 0.29).

The trochlear central trough was identified in each image as the point with the shortest perpendicular distance to the TEA. Then, the image with the trochlear central trough closest to the TEA was selected from each patient knee scan to calculate the trochlear medial asymmetry ratio and sulcus angle. To measure the trochlear medial asymmetry ratio, the trochlear central trough and medial/lateral aspects were projected onto the TEA to obtain the G–L and M–L lengths (Figure 1), and the ratio of G–L to M–L was calculated. In the healthy cohort, the mean predicted trochlear medial asymmetry ratio was 0.52, while the mean ground truth ratio was 0.51. Similarly, in the KA cohort, the mean predicted trochlear medial asymmetry ratio was 0.51, while the mean ground truth ratio was 0.51. There was no statistically significant difference between the ground truth and deep learning model predictions of trochlear medial asymmetry ratio in both the health and KA cohorts (*p* = 0.18 and 0.75, respectively). Additionally, there was no statistically significant difference in the ground truth trochlear medial asymmetry ratio between the healthy and KA cohorts (*p* = 0.86).

The sulcus angle was measured by computing the angle between the lines connecting the trochlear medial and lateral aspects to the central trough. In the healthy cohort, the mean predicted sulcus angle was 130.49°, while the mean ground truth sulcus angle was 127.19°. Similarly, in the KA cohort, the mean predicted sulcus angle was 121.90°, while the mean ground truth sulcus angle was 122.38°. Here, there was a statistically significant difference between the ground truth and predicted sulcus angle in the healthy cohort (*p* < 0.05). There was no statistically significant difference between the ground truth and predicted sulcus angle in the KA cohort (*p* = 0.78). Additionally, there was no statistically significant difference in the ground truth sulcus angle between the healthy and KA cohorts (*p* = 0.08).

Agreement of model predictions with ground truths for each patellofemoral parameter are presented in Bland–Altman plots with a ±1.96 standard deviation border to identify outliers. This analysis reveals few outliers in both the healthy and KA cohorts for all calculated parameters, demonstrating strong agreement between model predictions and ground truths for derived patellofemoral measurements (Figure 6).

## 4. Discussion

We have developed a deep learning model that accurately identifies key anatomic landmarks of the patellofemoral compartment on axial CT images of the distal femur within ~0.20–0.26 cm accuracy on a 512 × 512 original image and produces patellofemoral measurements with no statistically significant difference from human-derived measurements on both healthy and osteoarthritic knees. To our knowledge, this work represents the first deep learning regression model for automated cross-sectional patellofemoral annotation trained on both physiologic and pathologic CT imaging datasets at this scale. Related work by Ye et al. and E et al. developed their models on datasets of *n* = 1418 and *n* = 1431 radiographic images, respectively [19,20], while our model was trained with a larger dataset of *n* = 14,652 CT images (3162 healthy, 11,490 osteoarthritic for KA). This novel deep learning model has the potential to further enhance our ability to automatically analyze anatomy of the patellofemoral compartment or any other musculoskeletal site at scale.

There is significant clinical and scientific motivation in studying patellofemoral anatomy to further our understanding of physiological anatomy and improve surgical outcomes. A notable body of work has focused on characterizing the trochlear groove, a key determinant of patellofemoral function and dysfunction [3,13,26,27]. The sulcus angle is an anatomical parameter that has been extensively studied to understand its effect on patella cartilage and bone volume [28], its morphological differences between sexes [4,6,7] and ethnicities [7], and its role in trochlear dysplasia [8]. Other patellofemoral parameters have been characterized such as the “turning point” of trochlear groove orientation [1,2], medial/lateral trochlear facet width [8], and various lower limb angles (e.g., trochlear groove-posterior condylar angle) [5].

Related work studies the ability of TKA implants to replicate this physiological patellofemoral anatomy. Unfortunately, TKA implants often fail to restore or reproduce physiological patellofemoral trochlear groove geometry and many may even exhibit signs of trochlear dysplasia [10]. Rosa et al. conducted a morphological analysis of 4116 preoperative OA knees from the general Australian population and 45 TKA implant designs, and they found a significant mismatch between native and prosthetic implant trochlear angles [9]. Li et al. analyzed 523 knees from French (264 knees) and Chinese (259 knees) patients alongside 9 TKA implant designs, and they found that prosthetic trochlear were too shallow relative to French knees but within the 3rd quartile of Chinese knees [7]. Additional work studied implant positioning effects by segmenting physiological and OA knee imaging to produce 3D patellofemoral models and used analysis software to virtually compare native and prosthetic trochlear anatomy [11,12].

Our model has the potential to greatly expand the scope of this past work by offering the capability to deploy standardized and automated patellofemoral measurements in larger and more representative or diverse patient populations. Characterization of patellofemoral anatomy has previously been limited by the availability of high-quality, annotated imaging data, resulting in most studies having cohort sizes of *n* < 300 and a majority with *n* < 100. Larger scale studies can offer a more comprehensive understanding of patellofemoral anatomy in populations but are often limited by cohort representativeness. For example, studies by Koh et al. [6] and Rosa/Hazratwala [5,9] used *n* = 975 and *n* = 4116 subjects but were limited to Korean and Australian patient subpopulations, respectively. Patellofemoral imaging data are abundant, and with an automated measurement pipeline enabled by our deep learning model, it is possible to conduct morphological analyses on massive cohort sizes and enhance our understanding of physiological anatomy across various demographic dimensions such as gender, age, and ethnicity. At the clinical and patient level, our model enables patient-specific preoperative planning with measurements performed on cross-sectional imaging in a consistent, repeatable, and reliable fashion. Implications are significant for surgical planning and replication of patient native anatomy, in addition to future patient-specific implant design.

Our study also has limitations. During data collection, our radiology residents selected ranges of CT slices containing all seven landmarks. Thus, our model assumes that all incoming images contain all seven landmarks, so it is possible that images with partial or unclear trochlear and patellofemoral structures would yield poor predictions. Additionally, the imaging protocols for the KA cohort had a thinner slice thickness and produced more images compared to the healthy cohort, so the patch models were trained on more image patterns with OA conditions. This imbalanced dataset may have affected model development. In our future work, we will address these limitations by developing a classification model to automatically identify knee images containing entire trochlear and patellofemoral structures, and we will also include more healthy cohort images to solve the data imbalance.

Beyond the enhanced understanding of patellofemoral anatomy in populations, our model has the potential to enable novel research and clinical applications. Transfer learning is the process of adapting a model trained on one domain to a new application, and work by Zheng has demonstrated the effectiveness of using different imaging modalities (e.g., CT) to improve segmentation accuracy on a target modality (e.g., MRI) [29]. We believe that our CT-based model can be used in transfer learning to develop models in the MRI domain, which would enable the analysis of additional patellofemoral soft-tissue characteristics such as cartilage volume. Similarly, our model approach could be adapted to characterize other complex joints that demonstrate high population variability, such as the elbow. Clinically, our model could be further utilized to investigate associations between trochlear anatomical differences and TKA surgical outcomes in prospective and retrospective large-scale clinical studies. For example, complication and dissatisfaction rates (e.g., anterior knee pain) may be as high as 20% following TKA procedures [30]. Our model could be applied to help us better understand if anatomical differences play a role in the prevalence of these postoperative symptoms.

## 5. Conclusions

In this study, we developed a modified ResNet50 deep learning model with self-supervised pretrained weights to automatically detect 7 patellofemoral anatomic landmarks on axial slices of the distal femur and measure key patellofemoral parameters. The model was trained and evaluated using a dataset of 14,652 CT images from 483 patients across 6 sites acquired between April 2017 and May 2022. The dataset consisted of 2 cohorts: a healthy cohort with 206 asymptomatic patients, and an osteoarthritic cohort with 277 patients scheduled for total knee arthroplasty procedures. Ground truth landmark labels were annotated by trainees with final review and correction by a senior musculoskeletal radiologist.

The model demonstrated highly accurate landmark detection, with a mean absolute error (i.e., spatial distance) from ground truth annotations of 0.20 cm in the healthy cohort and 0.26 cm in the KA cohort. The model’s predictions for the patellofemoral measurements of TEA length, TEA-PFA angle, trochlear medial asymmetry ratio, and sulcus angle were compared to ground truth annotations. Statistical analysis showed no statistically significant difference between the model’s predictions and the ground truth measurements for TEA length, TEA-PFA angle, and trochlear medial asymmetry ratio in both cohorts. However, there was a statistically significant difference in the predicted sulcus angle compared to the ground truth in the healthy cohort, but not in the KA cohort. Bland–Altman plots demonstrated strong agreement between the model’s predictions and ground truths for all calculated parameters, with few outliers.

These initial results represent the first deep learning regression model for automated cross-sectional patellofemoral annotation trained on a large dataset of both healthy and osteoarthritic CT images. Studying patellofemoral anatomy is crucial for understanding its function and improving surgical outcomes, and previous research has focused on characterizing the trochlear groove and other patellofemoral measurements. Unfortunately, KA implants often fail to replicate this native patellofemoral anatomy, leading to implant complications and potential dysplasia. Our model enables standardized and automated measurements of the patellofemoral compartment, potentially expanding the scope of past studies by enabling the analysis of larger and more representative patient populations. This can enhance our understanding of patellofemoral anatomy across new demographic dimensions and enable patient-specific preoperative planning to improve KA and other surgical outcomes. Additionally, this model has the potential for transfer learning to analyze other imaging modalities and could be applied to other complex joints.

## Figures and Tables

**Figure 1 bioengineering-10-00815-f001:**
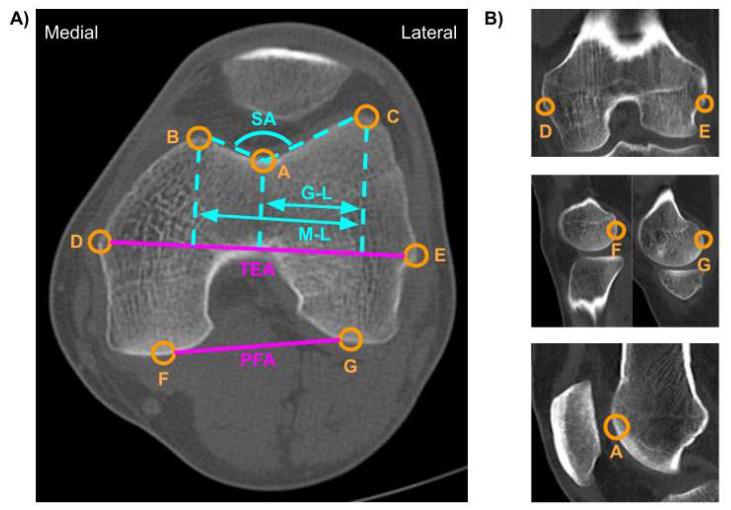
(**A**) Patellofemoral anatomical landmarks and measurements labeled on an axial knee CT. Anatomical landmarks are shown in orange: A: trochlear central trough; B and C: trochlear medial and lateral aspects, respectively; D and E: medial and lateral peripheral epicondyle, respectively; F and G: medial and lateral posterior condyle, respectively. Shown in purple: transepicondylar axis (TEA) and posterior femur axis (PFA). Shown in cyan: sulcus angle (SA), medial–lateral aspect length (M–L), and groove–lateral aspect length (G–L). Points B, A, and C define the femoral trochlear groove. (**B**) Anatomical landmarks shown on additional coronal and sagittal CT knee views.

**Figure 2 bioengineering-10-00815-f002:**
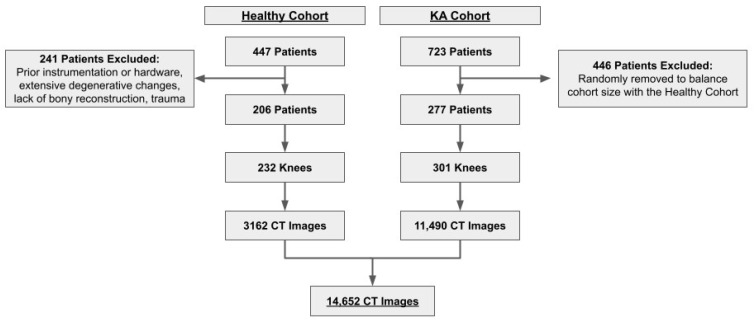
Flowchart of patient data set selection.

**Figure 3 bioengineering-10-00815-f003:**
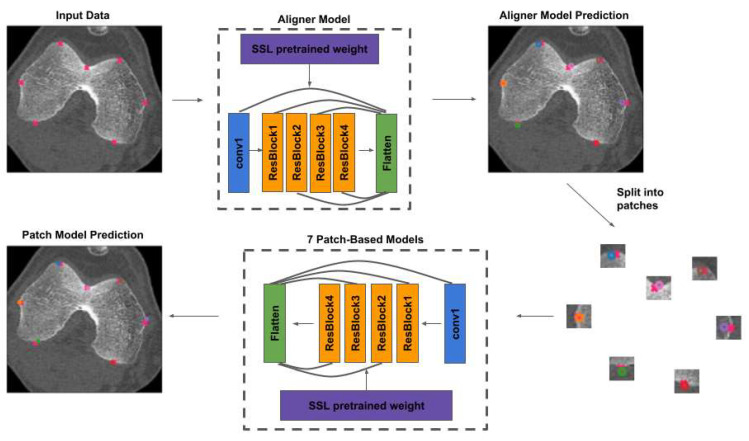
Two-stage patellofemoral landmark prediction model. The “aligner” model makes an initial prediction (circle without color fill) of all seven landmarks on a full image, and then seven “patch” models, one for each patellofemoral landmark, further refine the predictions (circle with color fill). Each color corresponds to one of the seven patellofemoral landmarks. A modified ResNet50 network was used as the backbone for all models, with a supervision mechanism that connects the last layer of each residual block to the final flattened layer. All models were initialized with self-supervised learning (SSL) pretrained weights and trained with the mean squared error loss function and Adam optimizer.

**Figure 4 bioengineering-10-00815-f004:**
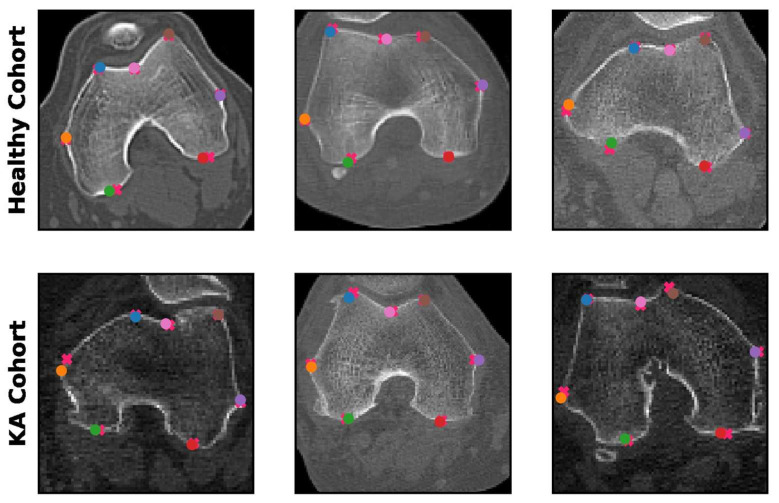
Landmark predictions. Human landmark annotations are shown with an “x” mark, while model landmark predictions are labeled with transparent “o” marks. Each color corresponds to one of the seven patellofemoral landmarks.

**Figure 5 bioengineering-10-00815-f005:**
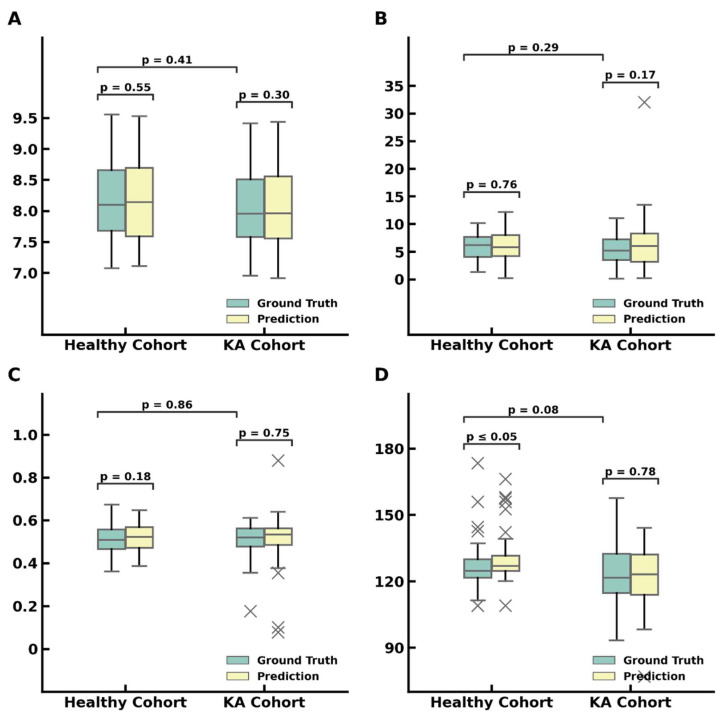
Patellofemoral measurements compared between the model predictions and ground truth annotations on the test dataset (*n* = 49 knees from the healthy cohort; *n* = 56 knees from the KA cohort) for: (**A**) TEA length, (**B**) TEA-PFA angle, (**C**) trochlear medial asymmetry ratio, (**D**) sulcus angle.

**Figure 6 bioengineering-10-00815-f006:**
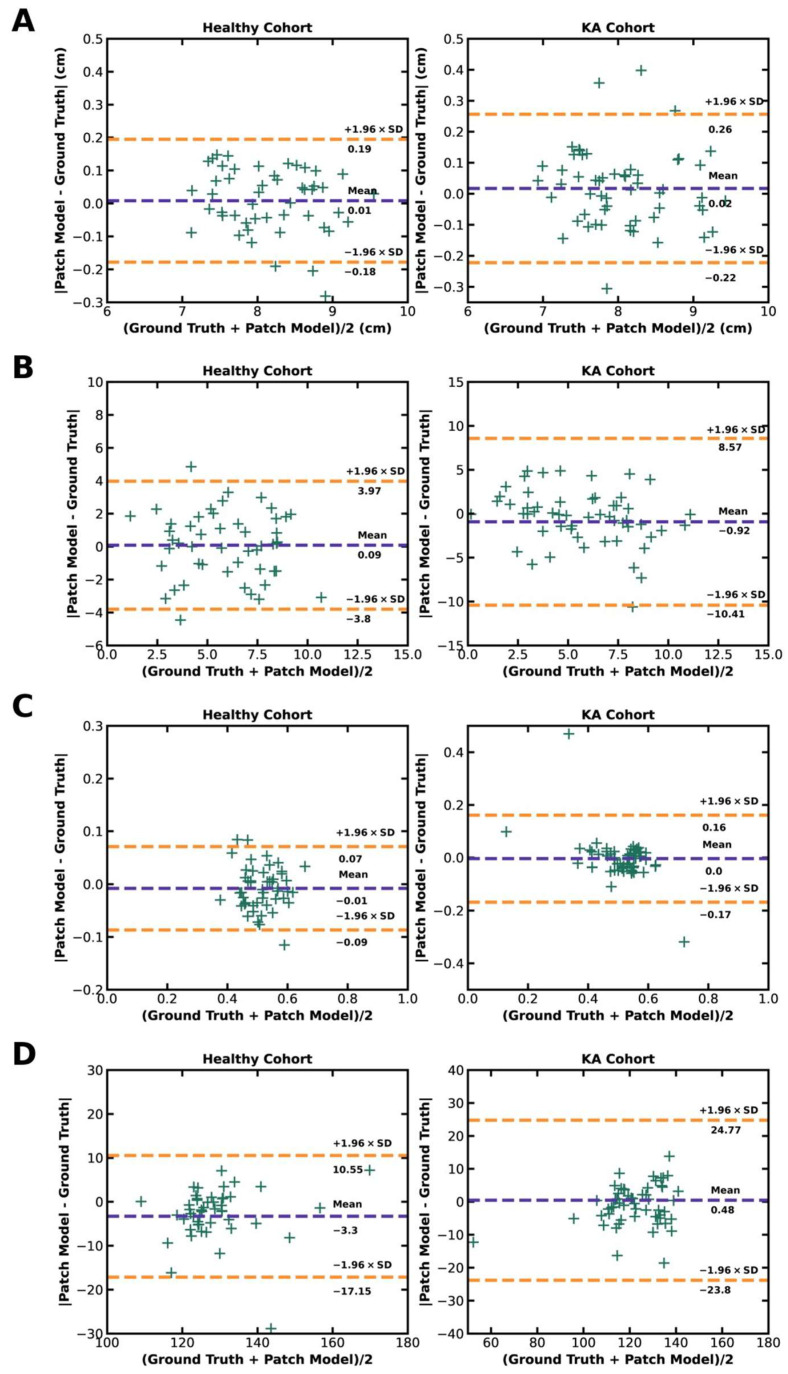
Bland-Altman plots of patellofemoral parameters on the test dataset of the healthy cohort (*n* = 49 trochlea) and the KA cohort (*n* = 56 trochlea): (**A**) TEA length, (**B**) TEA-PFA angle, (**C**) trochlear medial asymmetry ratio, (**D**) sulcus angle. “+” markers indicate each knee; the purple dotted line shows the mean of the absolute difference between the ground truth and prediction, while the two orange dotted lines show outliers with a 1.96 × standard deviation threshold.

**Table 1 bioengineering-10-00815-t001:** Summary of patient data characteristics by cohort.

	Healthy Cohort	KA Cohort	Overall Cohort
No. of patients	206	277	483
Age (IQ range)	33 (27–36.75)	65 (58–71)	53 (34–66)
Female/male	73/133	173/104	246/237

**Table 2 bioengineering-10-00815-t002:** Average Euclidean distance between predicted and ground truth landmarks and precision analysis (% in parenthesis) with a 0.40 cm distance threshold on the test dataset (*n* = 95 patients/105 knees) from the aligner model and patch models.

	Aligner Model	Patch Models	*p*-Value
Healthy cohort	0.42 cm (57.8%)	0.20 cm (97.3%)	<0.001
KA cohort	0.50 cm (50.3%)	0.26 cm (93.6%)	<0.001

**Table 3 bioengineering-10-00815-t003:** Summary of annotated versus model-predicted patellofemoral measurements on the test dataset (*n* = 95 patients/105 knees with 2872 images). ±values are standard deviation.

Patellofemoral Parameter	Ground Truth (Human Annotation)	Model Prediction	*p*-Value
TEA length			
Healthy cohort	8.17 ± 0.60 cm	8.16 ± 0.62 cm	0.55
KA cohort	8.07 ± 0.64 cm	8.06 ± 0.65 cm	0.3
TEA-PFA angle			
Healthy cohort	5.96 ± 2.38°	5.87 ± 2.49°	0.76
KA cohort	5.44 ± 2.62°	6.36 ± 4.85°	0.17
Trochlear medial asymmetry ratio			
Healthy cohort	0.51 ± 0.06	0.52 ± 0.06	0.18
KA cohort	0.51 ± 0.07	0.51 ± 0.11	0.75
Sulcus angle			
Healthy cohort	127.19 ± 10.69°	130.49 ± 10.86°	<0.05
KA cohort	122.38 ± 15.87°	121.90 ± 14.86°	0.78

## Data Availability

Data is unavailable due to privacy or ethical restrictions.

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
