# Peer review of "Deep Learning for Automated Measurement of Patellofemoral Anatomic Landmarks"

_bioengineering, 2023, doi:10.3390/bioengineering10070815_

Round 1
Reviewer 1 Report
The current article titled “Deep Learning for Automated Measurement of Patellofemoral Anatomic Landmarks” Ref: bioengineering-2481154, deals with an important subject. It is well written with understandable language.
A separate conclusion section is needed, explaining the entire study starting from the data used, analysis of the observations and final decision. So, the last paragraph of this study (page 13) should be revised intensively.
Reviewer 2 Report
The article showcases an intriguing study that employs a modified ResNet50 deep learning model to automatically measure knee anatomy, with specific focus on the patellofemoral joint, through CT images sourced from 483 patients. The model adeptly identifies crucial trochlear landmarks and delivers human-equivalent measurements on both healthy and pathological knees. The research effectively highlights the potential of deep learning for automated patellofemoral annotation, enhancing the scalability of our anatomical analysis. Here are my detailed comments:
· The introduction would benefit from an expanded explanation regarding the clinical significance of these measurements and their associated outcomes.
· Figure 1 should include a comprehensive view of knee anatomy, supplemented by various CT image views with marked corresponding areas.
· On Line 73, "inter-intra variability" should be the correct terminology.
· In Table 1, the presentation of sex data is somewhat confusing; it would be clearer to organize this as Female/Male rows and display the numbers for each cohort and overall.
· Briefly touching upon the inter-intra variability for annotators could add value.
· Could you clarify whether your algorithm predominantly identifies landmarks at a specific level where all landmarks are present? Please illustrate how your model operates during mismatch occurrences, potentially by demonstrating a mismatch on an image.
· Table 2 lacks a clear explanation; please include measurement units and specify the number of cases represented. A more organized structure for this table would be beneficial.
· Could you elaborate on why the KA cohort contains more images, given that the thickness of all acquired images is uniform? (Figure 2)
· Did an expert select the CT slice range containing trochlear groove geometry for all cases? Were any annotations for the anatomical landmark conducted by the expert? Could you provide details on how you evaluated the trainee's performance?
· Please provide additional details about the model's performance, including accuracy, sensitivity, specificity, precision, recall, F1 score, and AUROC. Define the pixel distance threshold that qualifies a prediction as accurate and precise.
· It would be informative to see a performance comparison between your “aligner” and “path” models, as it's unclear what benefits were derived from the second model.
· Figure 4 should be more zoomed-in and include varied image levels. Currently, it doesn't sufficiently illustrate the differences between healthy and diseased cases.
· For Figure 5, could you specify the number of data cases being analyzed? This image should also be referenced in the text.
· In Figure 6's explanation, please clarify why the outliers occurred. Were these random, or was there a specific pattern?
· Line 292 suggests that your model comprises more data. Could you detail in how many of your images all 7 desired locations were simultaneously present? As noted, a deeper dive into performance is necessary for better results, not solely relying on the quantity of samples. A power analysis for your cohort could also be insightful here.
· On Line 347, please specify which TKA surgical outcomes could be examined using your model.
I think the quality of the English language used in the paper was fine and understandable.
Reviewer 3 Report
The study entitled the Deep Learning for Automated Measurement of Patellofemoral Anatomic Landmarks, this study proposed was to hypothesize that a deep learning model with self-supervised pretrained weights could be trained to automatically detect patellofemoral anatomic landmarks. The main weakness of the study regards the novelty, I'm not sure what is being added to the literature.
There are already studies showing the relation between knowledge about this questions as (to name ones):
Automatic measurement of the patellofemoral joint parameters in the Laurin view: a deep learning–based approach https://pubmed.ncbi.nlm.nih.gov/35788755/
Future Directions in Patellofemoral Imaging and 3D Modeling https://www.ncbi.nlm.nih.gov/pmc/articles/PMC9076796/
Automatic Detection of Anatomical Landmarks on the Knee Joint Using MRI Data https://onlinelibrary.wiley.com/doi/pdf/10.1002/jmri.24516
Other issues:
Material and Methods: In this section, you need to clearly describe how individuals were approached, how many were approached, how many were eligible, consented or refused. Also, Inclusion and exclusion criteria should be cited with references and cited guidelines for this study may be recommended in order to improve the quality of the manuscript.
The Discussion section is a rehashing of the results. It does not appear that the authors include much interpretation of what the study findings mean for clinical practice or research.
FInally, the conclusión is weak and too long.
I consider that the study is not ready for publication and regret that the disposition is not favorable, but would like to thank you for your support.
I wish you all the best.
The study entitled the Deep Learning for Automated Measurement of Patellofemoral Anatomic Landmarks, this study proposed was to hypothesize that a deep learning model with self-supervised pretrained weights could be trained to automatically detect patellofemoral anatomic landmarks. The main weakness of the study regards the novelty, I'm not sure what is being added to the literature.
There are already studies showing the relation between knowledge about this questions as (to name ones):
Automatic measurement of the patellofemoral joint parameters in the Laurin view: a deep learning–based approach https://pubmed.ncbi.nlm.nih.gov/35788755/
Future Directions in Patellofemoral Imaging and 3D Modeling https://www.ncbi.nlm.nih.gov/pmc/articles/PMC9076796/
Automatic Detection of Anatomical Landmarks on the Knee Joint Using MRI Data https://onlinelibrary.wiley.com/doi/pdf/10.1002/jmri.24516
Other issues:
Material and Methods: In this section, you need to clearly describe how individuals were approached, how many were approached, how many were eligible, consented or refused. Also, Inclusion and exclusion criteria should be cited with references and cited guidelines for this study may be recommended in order to improve the quality of the manuscript.
The Discussion section is a rehashing of the results. It does not appear that the authors include much interpretation of what the study findings mean for clinical practice or research.
FInally, the conclusión is weak and too long.
I consider that the study is not ready for publication and regret that the disposition is not favorable, but would like to thank you for your support.
I wish you all the best.
Reviewer 4 Report
· The paper titled “Deep Learning for Automated Measurement of Patellofemoral Anatomic Landmarks” states that Patellofemoral joint stability is a complex problem with various bony and soft tissue contributors with substantial research focused on elucidating the patellofemoral parameters.
· Studies on the characterization of shape, position, and orientation of the trochlear groove have been conducted to study the patella positions during the range of knee motion.
· Characteristic features related to gender and ethnicity and the investigation of knee implants to replicate physiologic patellofemoral anatomy in both vivo and ex-vivo.
· There is interest from clinical viewpoint in understanding further the anatomy of patellofemoral compartment that enable the measurement of various parameters that quantify morphological characteristics (Figure 1).
· The use of deep learning has shown remarkable utility in detecting landmarks and objects in images over the last decade for studying medical and identity-related features, suggesting that deep learning can be equally applied to study trochlear groove geometry.
· Deep learning techniques can also be applied to medical imaging, particularly for patellofemoral characterization including automated measurement of various knee parameters, for example, detecting patellofemoral joint landmarks using axial radiography in the Laurin view.
· The success of these studies suggests that deep learning can be practically applied to study trochlear groove geometry by acquiring Computed tomography (CT) images of the knee from 483 patients were acquired between April 2017 through May 2022 and used in this study as shown in Figure 2 the flow chart of patient data collection and summary of patient data characteristics by cohort is listed in Table 1.
· All patellofemoral parameters are calculated based on the predicted coordinates, with performance of the patch-based model was evaluated by comparing the coordinates of the predicted trochlear landmarks with the ground truth annotation coordinates. The mean absolute error (i.e., spatial distance) between the prediction and the ground truth landmarks was 3.70 pixels in the Healthy Cohort and 5.33 pixels in the KA Cohort at the original 512x512 image resolution. This indicated highly accurate automated landmark detection comparable to human readers. Examples of predictions made by the patch-based model were presented in Figure 4.
Is the subject matter presented in a comprehensive manner?
· The 14-page paper is based on experimental details for 14654 images of 483 patients, discussing relevant concepts and results to support the “Deep Learning for Automated Measurement of Patellofemoral Anatomic Landmarks” genuinely.
· Table 2 (not referred to in the text} shows a Summary of annotated versus model-predicted patellofemoral measurements. ± Values are standard deviation with Landmark predictions. Human landmark annotations are shown with an “x” mark, while model landmark predictions are labeled with transparent “o” marks shown in Figure 4.
· Figure 5 shows Patellofemoral measurements compared between the model predictions and ground truth annotations for: A) TEA length, B) TEA/PFA angle, C) trochlear medial asymmetry ratio, D) sulcus angle.
Are the references provided applicable and sufficient?
· The authors take support from Twenty Six(26) some branded, and some old unbranded and including none from MDPI energies. The results are related and should be cross validated against similar recently reported work
· It is a good paper and support the title well with utility manufacturing readers. I decide to accept the paper for publication in the MDPI Bioengineering
Query 1. I suggest to replace references 3, 10, 12, and 17 be replaced by branded and current relevant references from MDPI bioengineering

· English language is understandable and one would to read the paper to support the title well with utility for readers in the medical imaging domain.
Round 2
Reviewer 3 Report
My opinion about the article remains the same of the first revision of manuscript. Most of the issues that I advanced to you cannot was repaired. A new study would be needed to make these things suitable. The clarifications provided do not solve the problem. Best regards.
My opinion about the article remains the same of the first revision of manuscript. Most of the issues that I advanced to you cannot was repaired. A new study would be needed to make these things suitable. The clarifications provided do not solve the problem. Best regards.